# Synthesis of a Cyclooctapeptide, Cyclopurpuracin, and Evaluation of Its Antimicrobial Activity

**DOI:** 10.3390/molecules28124779

**Published:** 2023-06-15

**Authors:** Rani Maharani, Hasna Noer Agus Yayat, Ace Tatang Hidayat, Jamaludin Al Anshori, Dadan Sumiarsa, Kindi Farabi, Tri Mayanti, Desi Harneti, Unang Supratman

**Affiliations:** 1Department of Chemistry, Faculty of Mathematics and Natural Sciences, Universitas Padjadjaran, Jatinangor 45363, West Java, Indonesia; hasna17008@mail.unpad.ac.id (H.N.A.Y.); ace.hidayat@unpad.ac.id (A.T.H.); jamaludin.al.anshori@unpad.ac.id (J.A.A.); dadan.sumiarsa@unpad.ac.id (D.S.); kindi.farabi@unpad.ac.id (K.F.); t.mayanti@unpad.ac.id (T.M.); nurlelasari@unpad.ac.id (N.); desi.harneti@unpad.ac.id (D.H.); unang.supratman@unpad.ac.id (U.S.); 2Laboratorium Sentral, Universitas Padjadjaran, Jatinangor 45363, West Java, Indonesia; 3Centre of Natural Products and Synthesis Studies, Faculty of Mathematics and Natural Sciences, Universitas Padjadjaran, Jatinangor 45363, West Java, Indonesia

**Keywords:** cyclopurpuracin, cyclooctapeptide, solid-phase peptide synthesis, cyclisation, antimicrobial peptide

## Abstract

Cyclopurpuracin is a cyclooctapeptide isolated from the methanol extract of *Annona purpurea* seeds with a sequence of cyclo-Gly-Phe-Ile-Gly-Ser-Pro-Val-Pro. In our previous study, the cyclisation of linear cyclopurpuracin was problematic; however, the reversed version was successfully cyclised even though the NMR spectra revealed the presence of a mixture of conformers. Herein, we report the successful synthesis of cyclopurpuracin using a combination of solid- and solution-phase synthetic methods. Initially, two precursors of cyclopurpuracin were prepared, precursor linear A (NH_2_-Gly-Phe-Ile-Gly-Ser(*t*-Bu)-Pro-Val-Pro-OH) and precursor linear B (NH-Pro-Gly-Phe-Ile-Gly-Ser(*t*-Bu)-Pro-Val-OH, and various coupling reagents and solvents were trialled to achieve successful synthesis. The final product was obtained when precursors A and B were cyclised using the PyBOP/NaCl method, resulting in a cyclic product with overall yields of 3.2% and 3.6%, respectively. The synthetic products were characterised by HR-ToF-MS, ^1^H-NMR, and ^13^C-NMR, showing similar NMR profiles to the isolated product from nature and no conformer mixture. The antimicrobial activity of cyclopurpuracin was also evaluated for the first time against *S. aureus*, *E. coli*, and *C. albicans*, showing weak activity with MIC values of 1000 µg/mL for both synthetic products, whereas the reversed cyclopurpuracin was more effective with an MIC of 500 µg/mL.

## 1. Introduction

Cyclic peptides isolated from animals, plants, bacteria, or fungi have been of interest due to their potential biological properties [1]. The unique and interesting biological properties of cyclic peptides have attracted many scientists and pharmaceutical companies to explore the compounds chemically and biologically [2]. Structurally, cyclic peptides are rigid, thereby reducing Gibbs free energy entropy, which improves their binding and selectivity to target molecules or receptors [3,4,5]. Several cyclic peptides were designed through side-chain-to-side-chain tethering to give an α-helix conformation to reduce conformational flexibility, hence improving the binding affinity and specificity to targets [6]. Furthermore, the cyclic peptides are more stable against protease and are resistant to exopeptidase hydrolysis due to the lack of both amino and carboxyl termini [7]. Several cyclic peptides from nature have been used clinically, such as gramicidin S, one of the oldest cyclic peptide drugs with strong antibacterial activity against gram-positive and gram-negative bacteria and some fungi. Other widely used cyclic peptide drugs include the antibiotics vancomycin and daptomycin, the immunosuppressant cyclosporine, and the hormones or hormone analogues octreotide, oxytocin, and vasopressin [8].

A new cyclic peptide of interest is cyclopurpuracin (**1**) (Figure 1), which was first isolated from *Annona purpurea* seeds by Gonzalez-Tepale and colleagues in 2018. Cyclopurpuracin is a cyclooctapeptide containing the amino acid sequence cyclo-Gly-Phe-Ile-Gly-Ser-Pro-Val-Pro, and all of these amino acids are L-configured (Figure 1) [9,10]. Information on the synthesis and biological properties of the compound has not been reported yet; therefore, the total synthesis of cyclopurpuracin (**1**) is of interest. The initial effort to synthesise cyclopurpuracin by our group was unsuccessful as the cyclisation step was problematic. Fortunately, reversed cyclopurpuracin (**2**) (Figure 1) was successfully synthesised using similar reaction conditions and was reported by Yayat et al. in 2022 [11]. 

A combination of solid- and solution-phase methods is commonly used for the synthesis of cyclic peptides. In this strategy, a linear precursor is prepared through solid-phase synthesis, while cyclisation is performed in a solution. This method was applied by Muhajir et al. in 2021 for the synthesis of nocardiotide A [12], Kurnia et al. in 2022 for the synthesis of xylapeptide B [13], Maharani et al. in 2021 for the synthesis of c-PLAI analogues [14], Zhang et al. in 2016 for the synthesis of dianthin I [15], and Rahmadani et al. in 2021 for the synthesis of exumolide A and B [16], as well as Yayat et al. in 2022 for the synthesis of reversed cyclopurpuracin, as mentioned earlier [11]. The latter strategy was applied in current studies.

Following the prior synthesis of reversed cyclopurpuracin, solid-phase linear precursor synthesis still takes advantage of the Fmoc/*t*-Bu-based strategy. Fmoc is the protecting group of the α amino group, while *t*-Bu is the protecting group of the side chain for the serine residue. The Fmoc group is base labile and the peptides are released from 2-chlorotrityl resin along with *t*-Bu deprotection using acid. Therefore, this orthogonal strategy ensures the peptide and all side-chain protecting groups remain on the resin during the elongation step [17].

In our initial studies published in Yayat et al. (2022), we investigated the cyclisation of the protected linear precursor of cyclopurpuracin. Linear precursors play an essential role in peptide cyclisation and determine whether cyclisation is straightforward. Linear precursors with eight amino acid residues are categorised as small linear peptides, which might be hard to make cyclic due to the flexible conformation [18,19]. In our present studies, several strategies were applied with the disconnection of cyclopurpuracin performed at two cyclisation sites. The first selected site was between proline as C-terminal and glycine as N-terminal to give a linear precursor A (NH_2_-Gly-Phe-Ile-Gly-Ser(*t*-Bu)-Pro-Val-Pro-OH) (Table 1). Proline at the C-terminus is essential because it suppresses any potential epimerisation that may occur during cyclisation and acts as a *β*-turn inducer to ease the cyclisation. The presence of glycine at the N-terminus makes cyclisation even easier due to its flexibility [20,21,22]. The second cyclisation site is between valine at the C-terminus and proline at the N-terminus to give a linear precursor B (NH-Pro-Gly-Phe-Ile-Gly-Ser(*t*-Bu)-Pro-Val-OH). The selection follows the cyclisation site of reversed cyclopurpuracin [11] to place proline at the N-terminus, which is reported to enhance the cyclisation efficiency and yield [23,24]. The presence of Gly and Pro in the middle sequences is also beneficial to bend the linear precursor. The hydroxyl group side chain of serine in the linear precursors is protected by *tert*-butyl. 

A head-to-tail cyclisation was applied in a highly diluted concentration to minimise any potential dimerisation or oligomerisation since there is an appropriate distance between molecules for an intramolecular reaction instead of an intermolecular interaction [20,25,26,27].

The selection of the coupling reagent for cyclisation is also important. Several reports employed HATU for cyclisation and obtained good results, such as those reported by Rahmadani et al. (2021) in the total synthesis of exumolides A and B [16] and Luo et al. (2018) in the synthesis of reniochalistatin E [28]. The cheaper HBTU reagent was also reported with good results, such as in the cyclisation of nocardiote A by Muhajir et al. in 2021 [12], NH_2_-Ser(*t*-Bu)-Asn-Leu-Ser(*t*-Bu)-Thr(*t*-Bu)-Asn-Val-Leu-OH by Gut et al. in 2001 [29], and cyclo-(L-Trp(Boc)-D-Leu-L-Lys(N_3_)-D-Leu-L-Trp(Boc)-D-Leu-L-Lys(N_3_)-D-Leu) by Chapman et al. in 2011 [30]. To avoid the formation of guanidine during cyclisation when using uranium reagents, as reported by Yang in 2015 [31], another reagent used in the present studies is a phosphonium-based coupling agent, PyBOP. Cyclisation using PyBOP has been reported in the cyclisation of samoamide A, tunicyclin D, brachystemin, F, reniochalistatin E, and [24,32,33,34]. Furthermore, another approach for peptide cyclisation takes advantage of the metal ion strategy (metal-ion-assisted cyclisation). The strategy was reported by Liu et al. in 2008, showing that the presence of metal univalent ions such as Na^+^ could promote cyclisation efficiency and shorten cyclisation times. The univalent ion could make an ion–dipole interaction with the oxygen of carbonyls; therefore the distance between the C-terminus and N-terminus is closer, achieving a beneficial conformation for cyclisation [35].

In this paper, we report our cyclisation studies to obtain cyclopurpuracin, which involved the selection of the cyclisation site and coupling reagents.

## 2. Results and Discussion

The linear precursors A and B were synthesised through a solid-phase method (Figure 1). The 2-chlorotrityl chloride resin (2-CTC) was employed as the solid support. The resin is often used in solid-phase peptide synthesis because it suppresses racemisation, avoids the formation of diketopiperazines, and provides a mild acidic cleavage condition [36]. The resin was initially swollen using dichloromethane to allow the reagent to access the polymer core. The first residue (Pro for linear precursor A and Val for linear precursor B) was attached to the resin with the addition of a DIPEA base in dichloromethane. The loading resin value of the first amino acid was 0.5 mmol/g for both linear precursors A and B, which was categorised as good (0.3–0.6 mmol/g) [37]. The following step was Fmoc deprotection using basic piperidine in dimethylformamide to give a free amino group that was ready to be reacted with the second amino acid. A combination of HBTU/HOBt as the coupling reagent with the presence of basic DIPEA in DMF was applied in all coupling reactions. A repetitive Fmoc deprotection and amide bond formation were undertaken to give the resin octapeptidyl-NH_2_ of precursors A and B. Once the linear octapeptide was constructed on the resin, the last step was to cleave the peptide from the resin by keeping the *t*-Bu protecting group on the backbone. The resin cleavage was performed employing a mixture of AcOH:TFE:DCM (2:2:6). The filtrate was collected and evaporated to obtain the protected linear octapeptide (**3**).

The crude linear precursors were purified by semi-preparative RP-HPLC [H_2_O: acetonitrile (95:5–20:80)] for 45 min. The chromatogram of linear precursor A showed a peak of the target compound at a retention time of 33.2 min, whereas the peak was at a retention time of 34.2 min for linear precursor B. All the fractions were collected and concentrated using a rotary evaporator to produce white solids (6.1 mg for linear precursor A and 12.2 mg for linear precursor B). The purified products were then characterised by HR-ToF-MS and the MS spectra showed correct molecular ion peaks of the desired linear octapeptide A and B at *m/z* [M + H]^+^ 829.4820 (calcd. 829.4820) and *m/z* [M + H]^+^ 829.4822 (calcd. 829.4824), respectively (Appendix A). Furthermore, the purity of the linear peptides was analysed by analytical RP-HPLC and the chromatograms showed a single peak at 10.7 min for linear precursor A and 10.6 min for linear precursor B as shown in Appendix A. The linear octapeptides were characterised by ^1^H-NMR and ^13^C-NMR (spectra can be found in Appendix A). The NMR spectra revealed the correct number of protons and carbons, showing that the spectral data are consistent with the synthetic products.

The cyclisation studies (Table 2) were performed based on the Muhajir et al. protocol in 2021, where the linear precursor A was dissolved in DCM to 1.25 mM with the addition of three equivalent HBTU and 1% *v/v* DIPEA at room temperature [12]. The reaction was monitored by TLC [*n*-hexane:ethyl acetate (6:4)] until there was no purple colour in the TLC plate when it was sprayed with ninhydrin solution. The reaction took seven days to yield the protected cyclopurpuracin and the remaining solvent was concentrated by a rotary evaporator. The crude peptide was then added by a deprotecting reagent (TFA:H_2_O:TIPS = 95:2.5:2.5) to eliminate the *t*-Bu protecting group. The cyclopurpuracin was successfully obtained and confirmed by HR-ToF-MS, showing a molecular ion peak at *m/z* [M+Na]^+^ 777.3809 (calcd. C_37_H_54_N_8_O_9_ 777.3808) (Appendix A). However, the HR-ToF-MS spectra indicated that the cyclic peptide was present as a minor trace; therefore, the coupling reagent was changed to HATU with the same reaction conditions. Unfortunately, the desired cyclic peptide was not formed after seven days, as confirmed by the presence of only one molecular ion peak of precursor linear A at *m/z* [M+H]^+^ 829.5936 (calcd. C_37_H_54_N_8_O_9_ 829.5935). The third trial cyclisation was performed using PyBOP but the desired product was still not formed after seven days with only one molecular ion peak of the linear precursor at *m/z* 829.4214 (calcd. C_37_H_54_N_8_O_9_ 829.4214). The failure of the cyclisation of linear precursor A may be due to the high flexibility of the linear octapeptide because of the presence of two glycine and one serine residues [38]. This flexibility maintains the distance between the C-terminus and N-terminus despite the presence of a proline residue that is known to restrict flexibility. 

Similar protocols were applied to cyclise the linear precursor B into cyclopurpuracin, but they were unsuccessful. The desired cyclic peptide and the precursor were not detected in the HR-ToF-MS spectra. It seems likely that the placement of valine, which is a β-branched residue, at the C-terminus did not favour cyclisation [26].

The cyclisation trial was continued using a protocol reported by Liu et al. in 2008 [35] that used NaCl to assist cyclisation. The linear precursors A and B were dissolved in DMF (2.0 mM) with the addition of PyBOP, DIPEA, and NaCl salt at room temperature. The reaction was monitored by TLC (*n*-hexane:ethyl acetate 6:4) and the ninhydrin test. Fortunately, the reaction only took two days to produce the protected cyclopurpuracin, as confirmed by HR-ToF-MS. The flexibility of the linear precursor is reduced because of the presence of Na^+^, which makes the C- and N-termini closer due to the ion–dipole interaction between the Na^+^ ion and the oxygen of each carbonyl (Figure 2), leading to a more straightforward formation of the cyclic product. 

Furthermore, the reaction mixture was extracted using a brine solution and ethyl acetate, and then the organic phase was concentrated by a rotary evaporator before the *t*-Bu deprotection step was performed using TFA:TIPS:H_2_O (95:2.5:2.5). The reaction mixture was then concentrated and the crude was characterised by HR-ToF-MS, showing the correct molecular ion peak at *m/z* 755.4087 (calcd. 755.4088) (Appendix A) for cyclopurpuracin obtained from linear precursor A (cyclopurpuracin A) and *m/z* 755.4069 (calcd. 755.4067) (Appendix A) for cyclopurpuracin obtained from linear precursor B (cyclopurpuracin B).

Crude peptide **1** was purified by semi-preparative RP-HPLC with a mobile phase (H_2_O: acetonitrile) (80:20–20:80) for 40 min, resulting in a white solid (3.8 mg for cyclopurpuracin A and 3.3 mg for cyclopurpuracin B). The purity was confirmed by analytical RP-HPLC with purified cyclopurpuracin A and B displaying single peaks at retention times of 12.7 min and 12.8 min, respectively (Appendix A). HR-ToF-MS spectra of purified **1** revealed a major peak at *m/z* [M + H]^+^ 755.4110 (calcd. C_37_H_54_N_8_O_9_ 755.4110) for cyclopurpuracin A and at *m/z* [M + Na]^+^ 777.3915 (calcd. C_37_H_54_N_8_O_9_ 777.3915) for cyclopurpuracin B, representing the correct molecular ion peak of the desired **1** (Appendix A and S14). Cyclic peptide **1** was further characterised by ^1^H-NMR and ^13^C-NMR (Appendix A). 

The structural assignment of cyclopurpuracin was performed using 1D NMR. The ^1^H NMR spectra of **1** showed 54 proton signals, including signals of secondary amide protons, aromatic protons, hydroxy protons, α-protons, aliphatic methines, aliphatic methylenes, and aliphatic methyls. The ^13^C NMR spectra showed 37 signals, including carbon signals of carbonyls, quarternary carbons, aromatic carbons, oxygenated carbons, α-carbons, aliphatic methine carbons, aliphatic methylene carbons, and aliphatic methyls carbons. The NMR spectra of cyclopurpuracin **1** showed the presence of a single conformer (Table 2), similar to the NMR spectra of the natural products of cyclopurpuracin [9]. This is different to the NMR spectra of reversed cyclopurpuracin reported by Yayat and colleagues in 2022, where it had two conformers [11], which is explained by the fact that cyclic peptides tend to show different conformations depending on the sequence that predominantly influences the peptide conformation. There is a hydrogen bond in the cyclopurpuracin between the hydroxy group of Ser with the carbonyl group of Pro that stabilises the structure and, eventually, a single conformer was detected in the NMR spectra [11]. This phenomenon was also found in the NMR spectra of AbE that revealed a single conformer due to the presence of a hydrogen bond between the hydroxy group in β-OHMePhe and the carbonyl group of Pro, which was different to the NMR spectra of AbA and [2*S*,3*S*-Hmp]-AbL, which showed two conformers [39,40]. 

The comparison of the ^1^H NMR and ^13^C NMR spectra of the synthetic cyclopurpuracin A and B to isolated cyclopurpuracin from nature revealed a high similarity to the chemical shifts of the natural products (Table 3).

The antimicrobial activity of cyclopurpuracin A and B and reversed cyclopurpuracin [11] were assessed against two bacteria and one fungus (Table 4). From the inhibition zone data, cyclopurpuracin A and B exhibited weak activity to inhibit the growth of gram-negative *E. coli*, gram-positive *S. aureus*, and *C. albicans* at a concentration of 1000 ppm. Strandberg and colleagues in 2021 [41] and Morales and colleagues in 2008 [42] described that the antimicrobial activities of a compound are categorised as strong when MIC values are <10 µg/mL, active when MIC values are <100 µg/mL, moderate when the MIC is in the range of 100–500 µg/mL, weak when the MIC values are in the range of 500–1000 µg/mL, and inactive when the MIC values are > 1000 µg/mL. In contrast, reversed cyclopurpuracin had a higher antimicrobial properties with MIC value of 500 µg/mL against all pathogens tested. It seems likely that the reversed sequence of a peptide could give different biological properties compared to the normal peptide sequence, as described previously [43,44].

## 3. Materials and Methods

### 3.1. Material 

The chemicals used in this research were 2-chlorotrityl chloride resin, dimethylformamide (DMF), dichloromethane (DCM), 2-(1H-benzotriazole-1-yl)-1,1,3,3-tetramethyluronium hexafluorophosphate (HBTU), O-(7-azabenzotriazole-1-yl)-*N,N,N′,N′*-tetramethyluronium hexafluorophosphate (HATU), benzotriazole-1-yloxytri(pyrrollidino)phosphonium hexafluorophosphate (PyBOP), *N*-hydroxybenzotriazole (HOBt), *n*-hexane *N*,*N*-diisopropyletilamine (DIPEA), piperidine, ethyl acetate, sodium chloride, 2,2,2-trifluoroethanol, acetic acid, triisopropylsilane, acetonitrile, sodium sulphate, and trifluoroacetic acid. All of the amino acid residues, Fmoc-L-Proline, Fmoc-L-Valine, Fmoc-L-Isoleucine, Fmoc-L-Serine(*t*-Bu)-OH, Fmoc-L-Glycine, Fmoc-L-Phenylalanine-OH, and 2-chlorotrityl chloride, were purchased from GL-biochem Ltd., Shanghai, China.

### 3.2. General Methods 

Analysis of the linear and cyclic octapeptide was achieved on the Waters 2998 Photodiode Array Detector (PDA) and LiChrospher 100 C-18 5 µm column RP-HPLC with a detection wavelength of 210 and 240 nm. The mobile phase consisted of acetonitrile (A) and deionised water (B) using gradient elution. The flow rate was 1.0 mL/min, and the column temperature was maintained at 25 °C over 40 min. The peptides were characterised by ^1^H-NMR and ^13^C-NMR on an Agilent NMR 500 MHz (^1^H) and 125 MHz (^13^C) using a deuterated solvent. Mass spectrometry spectra were recorded on the Waters HR-ToF-MS Lockspray. The loading resin absorbance was measured on the TECAN Infinite pro 200 UV-Vis Spectrophotometer.

### 3.3. General Procedure for the Synthesis of Linear Octapeptides, a Precursor of Cyclopurpuracin (2)

Chlorotrityl chloride resin was used as the solid support. First, 2-chlorotrityl chloride resin (400 mg, 0.6 mmol) was swollen in dichloromethane (10 mL) over 30 min at room temperature. The first amino acid (Fmoc-L-AA_1_-OH) (1 eq.) was loaded onto the resin in dichloromethane (4 mL) and basic DIPEA (2 eq.). For loading resin absorbance measurement, 20% piperidine in DMF (3 mL) was added to 0.6 mg of Fmoc-L-AA_1_ resin in an Eppendorf tube and left for 30 min followed by sonification for 5 min. The loading resin was measured by a UV-Vis spectrophotometer at 290 nm. For the next step, the capping of the resin was carried out by adding MeOH:DCM:DIPEA 15:80:5 (10 mL) twice for 15 min before the Fmoc group in Fmoc-L-AA_1_ resin was removed using 20% piperidine in DMF (5 mL) for 2 × 5 min to yield free amino groups. The free amino group was coupled with the second Fmoc-protected amino acid (Fmoc-AA_2_-OH) in the presence of HBTU (3 eq.) and HOBt (3 eq.) as a coupling agent and N, N-diisopropylethylamine (DIPEA) (6 eq.) as an activating reagent in DMF (4 mL) for 4 h at room temperature. The Fmoc group was removed using 20% piperidine in DMF (5 mL) for 2 × 5 min to afford the resin Fmoc-AA_1_-AA_2_-NH_2_. This cycle of coupling and deprotection was repeated with subsequent Fmoc-protected amino acids to afford the resin-coupled octapeptide. Finally, the resin was cleaved using AcOH:TFE:DCM (2:2:6) for 2 × 2 h. After the filtration of the resin and subsequent evaporation, the crude peptides were washed with dichloromethane repeatedly and dried under a vacuum. The linear peptide was analysed using analytical RP-HPLC (20–80% acetonitrile in water for 45 min, flow rate 1 mL/min, λ 240 nm), and the crude peptide was purified by semi-preparative RP-HPLC using 5–80% ACN eluent for 40 min. The target fractions (white powder, 6.1 mg for linear precursor A and 12.2 mg for precursor linear B) were collected and characterised by ToF-ESI-MS, ^1^H-NMR (500 MHz, CD_3_OD or DMSO-d_6_), and ^13^C-NMR (125 MHz, CD_3_OD or DMSO-d_6_).

Precursor linear A: white solid; 6.1 mg, yield: 15.2%; HR-TOF-MS *m/z* [M+H]^+^ 829.4820 (calcd. 829.4820). ^1^H-NMR (500 MHz, CD_3_OD, δ, ppm) 4.29 (1H, d, Pro^1^ Hα), 2.18; 2.12 (1H, m, Pro^1^ Hβ), 2.02; 1.90 (1H, m, Pro^1^ Hγ), 3.74; 3.54 (2H, m, Pro^1^ Hδ), 4.50 (1H, d, Val^2^ Hα), 3.10 (1H, m, Val^2^ Hβ), 0.87 (3H, d, Val^2^ CH_3_), 0.87 (3H, d, Val^2^ CH_3_), 4.54 (1H, d, Pro^3^ Hα), 2.28; 2.12 (1H, m, Pro^3^ Hβ), 2.02; 1.90 (1H, m, Pro^3^ Hγ), 3.74; 3.58 (1H, m, Pro^3^ Hδ), 4.78 (1H, d, Ser(*t*-Bu)^4^ Hα), 3.99; 3.38 (1H, d, Ser(*t*-Bu)^4^ Hβ), 1.16 (9H, s, Ser(*t*-Bu)^4^ CH_3_), 4.13 (2H, d, Gly^5^ Hα), 4.40 (1H, d, Ile^6^ Hα), 2.89 (1H, m, Ile^6^ Hβ), 1.11 (3H, d, Ile^6^ CH_3_-Hβ), 1.54 (2H, m, Ile^6^ Hγ), 0.93 (3H, d, Ile^6^ CH_3_-Hγ), 4.95 (1H, d, Phe^7^ Hα), 3.63; 3.18 (1H, d, Phe^7^ Hβ), 7.22 (2H, m, Phe^7^-2′,6′,3′,5′), 7.24 (1H, m, Phe^7^-4′), 3.83 (2H, d, Gly^8^ Hα). ^13^C-NMR (125 MHz, CD_3_OD, δ, ppm) 176.46 (Pro^1^ C=O), 62.1 (Pro^1^ Cα), 28.4 (Pro^1^ Cβ), 24.7 (Pro^1^ Cγ), 54.6 (Pro^1^ Cδ), 172.0 (Val^2^ C=O), 59.3 (Val^2^ Cα), 30.8 (Val^2^ Cβ), 18.5 (Val^2^ CH_3_), 176.45 (Pro^3^ C=O), 63.7 (Pro^3^ Cα), 29.3 (Pro^3^ Cβ), 24.5 (Pro^3^ Cγ), 52.0 (Pro^3^ Cδ), 167.6 (Ser^4^ C=O), 56.1 (Ser^4^ Cα), 61.6 (Ser^4^ Cβ), 28.2 (Ser^4^ *t*-Bu), 81.4 (Ser^4^ Cq-*t*Bu), 170.3 (Gly^5^ C=O), 42.0 (Gly^5^ Cα), 171.4 (Ile^6^ C=O), 60.4 (Ile^6^ Cα), 36.4 (Ile^6^ Cβ), 14.5 (Ile^6^ Cβ-CH_3_), 24.8 (Ile^6^ Cγ), 10.1 (Ile^6^ Cγ-CH_3_), 172.3 (Phe^7^ C=O), 58.5 (Phe^7^ Cα), 37.7 (Phe^7^ Cβ), 136.7 (Phe^7^ Cq), 128.2 (Phe^7^-2′,6′), 129.1 (Phe^7^-3′,5′), 126.5 (Phe^7^-4′), 165.0 (Gly^8^ C=O), 46.6 (Gly^8^ Cα).

Precursor linear B: white solid; 12.2 mg, yield: 20.3%; HR-TOF-MS *m/z* [M+H]^+^ 829.4822 (calcd. 829.4824). ^1^H-NMR (500 MHz, DMSO-d*_6_*, δ, ppm) 4.25 (1H, d, Val^1^ Hα), 1.75 (1H, m, Val^1^ Hβ), 0.78 (3H, d, Val^1^ CH_3_), 0.78 (3H, d, Val^1^ CH_3_), 7.97 (1H, d, Val^1^ NH), 4.37 (1H, d, Pro^2^ Hα), 2.74; 2.13 (1H, m, Pro^2^ Hβ), 2.02; 1.88 (1H, m, Pro^2^ Hγ), 3.60; 3.38 (1H, m, Pro^2^ Hδ), 4.59 (1H, d, Ser(*t*-Bu)^3^ Hα), 4.16; 3.91 (1H, d, Ser(*t*-Bu)^3^ Hβ), 1.05 (9H, s, Ser(*t*-Bu)^3^ CH_3_), 8.12 (1H, d, Ser(*t*-Bu)^3^ NH), 4.08 (2H, d, Gly^4^ Hα), 8.38 (1H, t, Gly^4^ NH), 4.25 (1H, d, Ile^5^ Hα), 2.78 (1H, m, Ile^5^ Hβ), 0.78 (3H, d, Ile^5^ CH_3_-Hβ), 1.55 (2H, m, Ile^5^ Hγ), 0.78 (3H, d, Ile^5^ CH_3_-Hγ), 7.50 (1H, d, Ile^5^ NH), 4.67 (1H, d, Phe^6^ Hα), 3.44; 3.19 (1H, d, Phe^6^ Hβ), 7.18 (2H, m, Phe^6^-2′,6′,3′,5′), 7.22 (1H, m, Phe^6^-4′), 8.17 (1H, d, Phe^6^ NH), 3.95 (2H, d, Gly^7^ Hα), 8.56 (1H, t, Gly^7^ NH), 3.68 (1H, d, Pro^8^ Hα), 1.88; 1.84 (1H, m, Pro^8^ Hβ), 1.70; 1.20 (1H, m, Pro^8^ Hγ), 3.04; 3.02 (1H, m, Pro^8^ Hδ). 

### 3.4. Cyclisation of Linear Octapeptide

#### 3.4.1. Using HATU/HBTU/PyBOP

The linear octapeptide was dissolved in dichloromethane in 1.25 mM before the addition of the coupling reagent (i.e., HBTU/HATU/PyBOP) (3 equiv.) and DIPEA (1% *v/v*). The mixture was stirred over 7 days at room temperature (monitored by TLC). The reaction mixture was concentrated to obtain a dark-yellow oil, which was deprotected with TFA:TIPS:H_2_O (95:2.5:2.5). 

#### 3.4.2. Using PyBOP/NaCl

The linear octapeptide was dissolved in DMF to 2.0 mM before the addition of PyBOP (1.14 equiv.) and DIPEA (1.65 equiv.). The mixture was stirred over 2 days at room temperature (monitored by TLC). The reaction mixture was extracted using a brine solution and ethyl acetate. The organic phase was added by Na_2_SO_4_ and then concentrated to yield a dark-yellow oil, which was deprotected with TFA:TIPS:H_2_O (95:2.5:2.5). The cyclooctapeptide was analysed using analytical RP-HPLC (20–80% acetonitrile in water for 45 min, flow rate 1 mL/min, λ 240 nm) and characterised by ToF-ESI-MS, ^1^H-NMR (500 MHz, DMSO-d_6_) and, ^13^C-NMR (125 MHz, DMSO-d_6_).

### 3.5. Microdilution Method

The cyclopurpuracin and the reversed cyclopurpuracin [11] were tested for their antibacterial activity using the microdilution method in a 96-well microplate with a Mueller–Hinton culture against three test pathogens (*S. aureus*, *E. coli*, *C. albicans*). The peptides were dissolved in 2% DMSO to a concentration of 1000 ppm. Each solution was dissolved gradually: 1000; 500; 250; 125; 62.5; 31.25; 15.62; 7.81; 3.90; 1.95; 0.97; and 0.48 ppm. The sample solutions, ciprofloxacin, vancomycin, nystatin, and 2% DMSO, were incubated in a 96-well microplate at 37 °C for 18 h. The plates were read in a spectrophotometer at a wavelength of 600 nm and the MIC was calculated as the percentage of microbial inhibition.

Cyclopurpuracin from precursor linear A: white solid; 3.8 mg, 20.8% yield; ^1^H-NMR (500 MHz, DMSO-d*_6_*, δ, ppm) and ^13^C-NMR (125 MHz, DMSO-d*_6_*, δ*,* ppm) data can be found in Table 1. HR-TOF-MS *m/z* [M + H]^+^ 755.4110 (calcd. C_37_H_54_N_8_O_9_ 755.4110). 

Cyclopurpuracin from precursor linear B: white solid; 3.3 mg, 18.1% yield; ^1^H-NMR (500 MHz, DMSO-d*_6_*, δ, ppm) and ^13^C-NMR (125 MHz, DMSO-d*_6_*, δ*,* ppm) data can be found in Table 1. HR-TOF-MS *m/z* [M + Na]^+^ 777.3915 (calcd. C_37_H_54_N_8_O_9_ 777.3915). 

## 4. Conclusions

Cyclopurpuracin (cyclo-Gly-Phe-Ile-Gly-Ser-Pro-Val-Pro) was synthesised through a combination of solid- and solution-phase methods using PyBOP with the addition of NaCl salt with linear precursor A (NH_2_-Gly-Phe-Ile-Gly-Ser(*t*-Bu)-Pro-Val-Pro-OH) and linear precursor B (NH-Pro-Gly-Phe-Ile-Gly-Ser(*t*-Bu)-Pro-Val-OH) in overall yields of 3.2% and 3.6%, respectively. The NMR spectra of cyclopurpuracin showed a single conformer and high similarity with natural cyclopurpuracin. The antimicrobial activity of cyclopurpuracin A and B was categorised as weak with MIC values of 1000 µg/mL and slightly lower than the antimicrobial properties of the reversed cyclopurpuracin with MIC values of 500 µg/mL.

## Data Availability

The data are contained within the article and Appendix A.

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
