# Peer review of "Synthesis of a Cyclooctapeptide, Cyclopurpuracin, and Evaluation of Its Antimicrobial Activity"

_molecules, 2023, doi:10.3390/molecules28124779_

Round 1

Reviewer 1 Report

The manuscript entitled “Synthesis of a cyclooctapeptide, cyclopurpuracin, and evaluation of its antimicrobial activityreported the synthesis of two cyclopurpuracin-derived cyclic peptides through the combination of solid- and solution-phase synthesis. Their antimicrobial activity was evaluated against S. aureus, E. coli, and C. albicans using a microdilution test.

Finally, I have to raise the following minor issues.

-       Introduction (Lines 37-41): In the literature, there are many cyclic antimicrobial peptides obtained through the application of several kind of cyclizations, including both head-to-tail and side chain-to-side chain cyclizations. The authors should describe these different approaches and add some references such as “Pharmaceutics. 2022 Feb 21;14(2):454. doi: 10.3390/pharmaceutics14020454”; Mar Drugs. 2022 Jun; 20(6): 397. doi: 10.3390/md20060397. These papers can support cyclization strategies as an approach to enhance antimicrobial activity.

-       Line 88: The authors state that glycine is the b-turn inductor similar to Proline but I think it is not completely correct. In fact, in the peptide chemistry, glycine is usually used as a spacer and it is a flexible amino acid. Please, should modify this sentence.

-       Introduction (Lines 77–120): The introduction usually describe the state of art of a work, while the authors describe in detail the obtained results. It is better to describe the state of the art on cyclopurpuracin and move these many information in the results section.

-       Results: The authors just reported the NMR results but it is not satisfying. The authors should improve this section by adding the peptide synthesis and Table 1 that they reported in the discussion. The authors should add the antimicrobial activity and the table with the MICs values in the results.

-       Discussion: Maybe, the authors should include the Results and discussion in one section because the format of the paper is a bit complicated to understand for the readers. Please, improve this aspect.

Author Response

Dear Reviewer,

Thank you for the comments for our manuscript. Herewith, I attached our response to the comments.

Thank you.

Regards,

Rani

Reviewer 2 Report

This research article by Rani Maharani and coworkers reported the synthesis of cyclooctapeptide and cyclopurpuracin by a combination of solid- and solution-phase synthetic methods from linear cyclopurpuracin. The cyclization product’s structure has been confirmed by HRMS and NMR. Importantly, the author conducted a microdilution test and the results showed that the cyclization products have weak activity with MIC values of 1000 µg/mL. The manuscript is well organized and written. I think the manuscript deserves to be published in Molecules after minor revision.

1)   Line 60: “Figure 2.” Should be “Figure 2.” (?), there is no Figure 1.

2)   From 140 lines: “DMSO-d6” should be “DMSO-d6”, same correction to 151-156 lines.

3)     Line 175: Please indicate the ratio of “AcOH:TFE:DCM”.

4)     Line 407, there is extra space.

5)     Line 421: “H2O” Should be “H2O”. 

6)  In the reference part, all of the references using full journal name, suggest using abbreviation.

Author Response

(The authors gave the same response as above.)

Reviewer 3 Report

This research is focused on the first total synthesis of cyclopurpuracin (cyclo-Gly-Phe-Ile-Gly-Ser-Pro-Val-Pro) through a combination of solid and solution-phase method. The topic is significantly original owing to the first synthesis of cyclopurpuracin A & B. Moreover, the synthesized compounds have been characterized by 1H-NMR, 13C-NMR  spectroscopy and mass spectrometry. The authors have also described the specific gap.

Overall, the formatting and description of methodology is well-reported. Authors have provided description about 1H-NMR and 13C-NMR spectroscopy. But relevant 1H-NMR and 13C-NMR spectras of cyclopurpuracin synthesized from precursor A & B need to be provided in supplementary information. Moreover, 13C-NMR spectrum of precursor B is missing in supplementary data.

·         Line 60, its not ‘’Figure 2’’. Correct the numbering.

·         In scheme 1, structures and arrows should be equally spaced throughout the manuscript.  It is better to keep some space between structure and each arrow to give a clean outlook.

·         Line 262, it should be Table 2. Recheck and update.

·         Line 307, it should be referred as ‘’Table 3’’.

·         Line 325, It should be labelled as ‘’Table 4’’.

 Other Remarks:

1-      Line 14, ‘’of cyclopurpuracin’’ should be omitted.

2-      Line 15, add space between ‘’eventhough’’

3-      Line 17 & 61, remove dash after solid.

4-      Line 17, add word ‘’by’’ before word ‘’using’’.

5-      Line 23, add and before ‘’13C-NMR’’

6-      Line 22, word ‘’products’’ should be replaced with ‘’product’’.

7-      Line 42, add comma after ‘’ gramicidin S’’

8-      Line 23, remove comma before word ‘’respectively’’.

9-      Line 47, add the word ‘’the’’ after ‘’one of’’.

10-   Line 70, add ‘’the’’ after word ‘’following’’

11-   Line 74 & 75 needs to be re-written.

12-   Line 77, add ‘’the’’ before ‘’cyclization’

13-   Line 69, modify the sentence.

14-   Sentence between the ‘’83-85’’ lines need to be re-written.

15-   Line 173, remove extra spacing.

16-   Line 157, remove extra spacing.

17-   Two consecutive paragraphs after scheme 1 are almost similar. Differentiate them.

18-   Line 104, result should be replaced with ‘’results’’.

19-   Line 110, add ‘’the’’ before’’cyclization of’’

20-   Line 146, there should be table 2.

21-   Line 230, add space before and after the equality sign.

22-   Line 234, replace word ‘’condition’’ with ‘’conditions’’.

23-   Line 245, add comma after ‘’flexibility’’.

24-   Line 258, remove dash after C.

25-   Line 266, remove extra space.

26-   Line 268 & 269 need to be re-written.

27-    Line 283, add ‘’and’’ before C-NMR.

28-   Line 407, remove extra space.

29-   Line 402, add space between number and degree.

30-   Re-write the sentence in line 379 & 380.

31-  Line 372, remove extra space.

English language needs to be carefully checked and improved.

Author Response

(The authors gave the same response as above.)

Round 2

Reviewer 3 Report

The authors have revised the manuscript and it can now be accepted in current form.